# Predicting Ischemic Stroke Patients to Transfer for Endovascular Thrombectomy Using Machine Learning: A Case Study

**DOI:** 10.3390/healthcare13121435

**Published:** 2025-06-16

**Authors:** Noreen Kamal, Joon-Ho Han, Simone Alim, Behzad Taeb, Abhishek Devpura, Shadi Aljendi, Judah Goldstein, Patrick T. Fok, Michael D. Hill, Joe Naoum-Sawaya, Elena Adela Cora

**Affiliations:** 1Department of Industrial Engineering, Dalhousie University, Halifax, NS B3H 4R2, Canada; joon-ho.han@dal.ca (J.-H.H.); abhishekdevpura@dal.ca (A.D.); shadi.aljendi@unb.ca (S.A.); 2Department of Community Health and Epidemiology, Dalhousie University, Halifax, NS B3H 4R2, Canada; 3Department of Medicine (Neurology), Dalhousie University, Halifax, NS B3H 3A7, Canada; 4Nova Scotia Health, Halifax, NS B3S 0H6, Canada; behzad.taeb@nshealth.ca (B.T.); adela.cora@nshealth.ca (E.A.C.); 5Department of Mathematics and Statistics, Dalhousie University, Halifax, NS B3H 3J5, Canada; simonealim@dal.ca; 6Faculty of Computer Science, University of New Brunswick, Fredericton, NB E3B 5A3, Canada; 7Department of Emergency Medicine, Dalhousie University, Halifax, NS B3H 4R2, Canada; judah.goldstein@emci.ca (J.G.); patrick.fok@nshealth.ca (P.T.F.); 8Emergency Health Services, Halifax, NS B3J 3A5, Canada; 9Department of Clinical Neurosciences, University of Calgary, Calgary, AB T2N 1N4, Canada; michael.hill@ucalgary.ca; 10Ivey School of Business, Western University, London, ON N6A 3K7, Canada; jnaoum-sawaya@ivey.ca; 11Department of Diagnostic Imaging, Dalhousie University, Halifax, NS B3K 6R8, Canada

**Keywords:** decision making, endovascular thrombectomy, EVT, ischemic stroke, machine learning, transfer

## Abstract

**Introduction**: Endovascular thrombectomy (EVT) is highly effective for ischemic stroke patients with a large vessel occlusion. EVT is typically only offered at urban hospitals; therefore, patients are transferred for EVT from hospitals that solely offer thrombolysis. There is uncertainly around patient selection for transfer, which results in a large number of futile transfers. Machine learning (ML) may be able to provide a model that better predicts patients to transfer for EVT. **Objective**: The objective of the study is to determine if ML can provide decision support to more accurately select patients to transfer for EVT. **Methods**: This is a retrospective study. Data from Nova Scotia, Canada from 1 January 2018 to 31 December 2022 was used. Four supervised binary classification ML algorithms were applied, as follows: logistic regression, decision tree, random forest, and support vector machine. We also applied an ensemble method using the results of these four classification algorithms. The data was split into 80% training and 20% testing, and five-fold cross-validation was employed. Missing data was accounted for by the k-nearest neighbour’s algorithm. Model performance was assessed using accuracy, the futile transfer rate, and the false negative rate. **Results**: A total of 5156 ischemic stroke patients were identified during the time period. After exclusions, a final dataset of 93 patients was obtained. The accuracy of logistic regression, decision tree, random forest, support vector machine, and ensemble models was 68%, 79%, 74%, 63%, and 68%, respectively. The futile transfer rate with random forest and decision tree was 0% and 18.9%, respectively, and the false negative rate was 5.37 and 4.3%, respectively **Conclusions**: ML models can potentially reduce futile transfer rates, but future studies with larger datasets are needed to validate this finding and generalize it to other systems.

## 1. Introduction

Stroke is a leading cause of mortality and morbidity [1]. Data from Nova Scotia shows that there are 125 ischemic stroke per 100,000 people in 2017 [2], and national data shows the morality due to ischemic stroke is 12.1% with 34.6% discharged home without services, 11.4% discharged home with services, 28.4% discharged to inpatient rehabilitation, and the remainder discharged to long-term care [3]. Fortunately, thrombolysis treatment for acute ischemic stroke was proven in 1995 [4], and the subsequent pooled analysis [5] showed the criticality of fast treatment and the need to treat with thrombolysis within 4.5 h with the odds of excellent outcomes with thrombolysis at 1.75 (1.35–2.27) when treated in less than 3 h; this falls to 1.15 (0.95–1.40) when treated after 4.5 h. Similarly, the mortality also increases with treatment delays with the odd ratio of death with thrombolysis of 1.00 (0.81–1.24) in less than 3 h to 1.22 (0.99–1.50) when treated after 4.5 h [5]. This evidence guided healthcare systems across Canada to create pre-hospital stroke bypass systems, where emergency medical services (EMS) bypass closer hospitals to take suspected stroke patients to hospitals capable of thrombolysis treatment [6]. In 2015, there was a significant leap forward in the treatment of the most severe ischemic stroke patients, as a series of randomized controlled trials proved endovascular thrombectomy (EVT) to be highly efficacious [7,8]. This treatment is provided to patients with a target large vessel occlusion (LVO); approximately 30–40% of all ischemic strokes are due to a LVO [9]. EVT is highly effective; approximately 26.9% of EVT treated stroke patients will recover with no disability compared to 12.9% of patients who did not receive EVT, or 46.0% of EVT treated patients will have only minor disability compared to 26.5% of patients who did not receive EVT [7]. The pooled analysis demonstrated that EVT is a highly efficacious treatment with a NNT (number needed to treat) of 2.6 for a reduction in disability or improving the modified Rankin scale score by 1 point at 90 days [7]. Thrombolysis treatment is synergistic to EVT; 75% of EVT patients in the EVT trials received thrombolysis, and the other 25% were not eligible to receive thrombolysis treatment. Unfortunately, EVT requires specialized equipment and personnel, which limits its availability to larger centers. Patients that are eligible for EVT but arrive at a hospital that is only capable of thrombolysis need to be urgently transferred to the EVT center for treatment. This creates significant hurdles to provide EVT to transferred patients, as not only does the efficacy of treatment decay with time [10] but eligibility for EVT treatment also decays with time [11].

Eligibility for EVT is primarily determined using imaging; yet despite this, there is uncertainty around how to best select patients for transfer from remote hospitals that are only capable of thrombolysis treatment to an EVT-capable center. A study from Ontario showed that 34% of stroke patients with LVO that were transferred for EVT from a peripheral site end up receiving EVT. Specifically, the futile transfers force the use of scarce ground ambulance and air transport resources for no benefit to the patient with 66% of those transferred for EVT being deemed ineligible for the treatment upon arrival [12]. Similar data from the US shows that only 27% of transferred patients received EVT [13]. While over-selecting patients for transfer to receive EVT may lead to a greater number of patients that will receive the treatment, this may also lead to a larger number of patients that turn out to be ineligible for treatment upon arrival. This essentially uses the philosophy of “casting a wide net”, which means that the stroke system transfers patients with even a small chance of receiving treatment; however, over-selection comes at a significant cost to the healthcare system due to the resources required for the urgent transport of a large proportion of severe stroke patients from a remote hospital to an EVT center for a patient who will ultimately not receive treatment, which is called a futile transfer. On the other hand, under-selection or being more discriminant in selection results in missed cases and an overall lower number of patients from remote hospitals that will receive EVT, but the overall cost of transfer is lower, since fewer patients are transferred.

The challenges with the decision to transport acute ischemic stroke patients for EVT can potentially be aided with the application of machine learning (ML). ML is a branch of artificial intelligence that builds data-driven statistical models to predict specific outcomes, which can be very helpful in healthcare systems; for example, ML can be used to provide decision support to clinicians where uncertainty and ambiguity exists, thus the ML models can be developed for healthcare systems to optimize patient outcomes. ML may be able to predict patients to transfer for EVT with information available at the time of decision making. ML is being increasingly used in stroke research. However, studies using ML have been limited to predicting outcomes based on patient demographic data, pre-morbidity, and stroke features, such as stroke severity [14,15,16,17]. Some studies have used univariate analysis to assess predictors for EVT eligibility for patients arriving at an EVT capable center [18,19] but not for transferred patients, and the effect of inter-hospital transfer as a single predictor of EVT eligibility has been studied from the receiving hospital perspective [20]. Additionally, the application of ML has been explored to predict outcomes for acute stroke patients generally [21,22,23,24,25], to forecast the outcome of EVT treatment [26,27,28,29], to interpret imaging in order to identify patients for treatment [30,31,32], to select older adults for EVT [33], and to help streamline the treatment process using in-hospital tracking system at an endovascular hospital [34].

In this study, we will apply ML to predict which acute ischemic stroke patients with an LVO should be transferred for EVT. The input variables will include information that is available at the time the transfer decision is made, and the response variable will indicate whether EVT was ultimately performed. The objective of the study is to determine whether ML can improve the accuracy of patient selection for EVT transfer.

## 2. Materials and Methods

### 2.1. Context

The province of Nova Scotia in Canada is used as a case study for this research, which has a single universal health system. Nova Scotia is a small Canadian province of approximately one million people and a land size of 52,942 km^2^. Nova Scotia has a significant coastline of approximately 7500 km, which makes road transport often much longer than air transport. Nova Scotia has one EVT-capable center and ten centers that are only capable of thrombolysis treatment. A map of Nova Scotia with all 10 thrombolysis-only centers and the single EVT-capable center is shown in the Appendix A.

### 2.2. Data Collection

All ischemic stroke patients from Nova Scotia who were admitted to a hospital from 1 January 2018 to 31 December 2022 were identified using the provincial stroke registry. From these patients, those that arrived at one of the 10 thrombolysis centers and were subsequently transferred to the EVT-capable center within 24 h of arrival were included. The variables from the registry used in this study are sex, age at time of stroke, time from onset of stroke to CT at thrombolysis-only center, and whether thrombolysis was administered. The time the patients left the thrombolysis-only center, and the modality of transfer (ground or helicopter) were obtained from the Nova Scotia Emergency Health Services (EHS), which is the single universal ambulance system for the province; the departure time was then used to obtain door-in-door-out time at the thrombolysis-only center. For these patients, the imaging from the thrombolysis-only centers were reread to obtain ASPECTS (Alberta Stroke Program Early CT Score), thrombus location, and collateral status. For collateral status, a radiologist read the multi-phase CTA images from the thrombolysis centres and assessed the collaterals as good, intermediate, or poor. We also obtained the driving distance and Euclidean distance between each thrombolysis-only center and the EVT-capable center, and for distance, we used the driving distance when the patient was transferred by ground ambulance and Euclidean distance when the patient was transferred by helicopter. The full list of variables is provided on Table 1.

Patients that did not have an LVO were removed from the dataset, as they were transported for a reason other than EVT. Furthermore, patients with a posterior stroke were excluded, thus limiting the study to anterior ischemic stroke patients. The reason for this is because the clinical trials for EVT were limited to anterior stroke patients, and there is less evidence for EVT in posterior circulation stroke patients [35,36]. Unfortunately, we could not include NIHSS, due to this variable being missing in the majority of records. The collection of NIHSS has been challenging to collect at smaller regional centres, but the importance of this measure is recognized, and this will be a limitation of this current study.

### 2.3. Data Analysis

Supervised binary classification ML algorithms were applied to the dataset, because the response variable of whether the patient received EVT is binary. The binary classification algorithms were applied to predict if the patient should be transferred for EVT. Four different supervised learning algorithms specializing in binary classification were applied along with an ensemble model to the preprocessed dataset, as follows: logistic regression, decision tree, random forest, and support vector machine (SVM). The ensemble model was built using a voting classifier that combines the four supervised models. These models account for data non-linearity, and they work reasonably well for small datasets that have a mix of numerical and categorical variables.

The data was split into training (80%) and testing (20%) with 5-fold cross-validation, which is a method that splits the dataset into 5 unique sets of 80–20 split. We train the model on each subset and use the remaining subsets as the testing set to evaluate the model. This process is repeated 5 times, and each subset is used as the testing set exactly once. Finally, the evaluation result is averaged over 5 trials [37].

Missing data was accounted by the k-nearest neighbour’s algorithm (k-NN), a multi-variate approach. It identifies k-nearest neighbours for a missing data point from all complete instances in a dataset by calculating the Euclidean distance between them. The Euclidean distance calculates a straight line between the missing point, and the other numbers are given by the following equation.(1)$$d\left(x,y\right)=\sqrt{\overset{n}{\underset{i=1}{\sum }}{\left|y_{i}-x_{i}\right|}^{2}}$$

The k value in k-NN represents the number of neighbouring points that are considered for missing value imputation. In the case of a categorical variable, the missing datum is filled with the most frequent value occurring in the neighbours. Whereas, for missing numerical variables, the missing information is imputed with the mean of the neighbours [38].

We performed hyperparameter tuning and balanced the class weights to prevent overfitting and biases and improve overall model performance. We used a grid search with a range of predefined values to find the optimal hyperparameters for all the models.

The accuracy of the models was determined by comparing the accuracy, ROC AUC (receiver operating characteristics area under the curve), F1 score, and precision for all the models. We also ran each patient data through the model to determine if the model indicated that the patient was a candidate for EVT transfer and compared this to the actual data. From this, the futile transfer rate was calculated for the model to compare with the actual data to determine if the model was able to reduce the futile transfer rate. A similar process was used to calculate the total rate of false negatives that the model outputted. The reason for a specific focus on false negative is because a false negative has the greatest negative impact in the use of the ML for decision making; specifically, a false-negative in this model means that the model is recommending that a patient is not transferred for EVT when they are potentially eligible for this treatment, which means that this would result in a poor patient outcome. A false positive, on the other hand, would not result in a poor patient outcome, as transferring a patient who would not be eligible for EVT results in the same patient outcome, but at a higher cost to the healthcare system due to the unnecessary transfer. False positives are taken into consideration in the F1 score and futile transfer rate. A feature importance analysis was completed for the best two performing models. All hyperparameters that were selected for this study are provided in the Appendix A. All analyses were performed with Python version 3.2 using the following libraries: pandas, numpy, sklearn, matplotlib, and random. Tests for normality were conducted using the Shapiro–Wilk test.

### 2.4. Ethical Approval

Ethical approval for this study was obtained from the Nova Scotia Health Research Ethics Board (File Number: 1028274).

## 3. Results

There were 5156 ischemic stroke patients identified in the Provincial Stroke Registry from 1 January 2018 to 31 December 2022. Out of these patients, 3103 arrived at one of the ten thrombolysis-only centers, and 239 of these patients were transferred to the EVT-capable center. There were 116 patients that were transferred within 24 h, and 23 patients were missing ASPECTS, where 18 were because the thrombus was in a posterior circulation vessel. The final dataset had 93 patients. The total futile transfer rate for the final 93 records was 49.46% (46 patients were transferred without receiving EVT). The data exclusion and inclusion are shown in Figure 1.

Descriptive statistics of the key parameters used in this study are provided in Table 2. The data for the final dataset had 77 records (83%) with an ASPECTS of 8 to 10. The most common occlusion location was M1 with 37 records (40%) followed by 27 (29%) with a tandem or tandem ICA occlusion; there were 15 records (16%) with an M2 occlusion. Most of the patients had good collaterals 51 (55%), while only 8 (9%) had poor collaterals. There were 10 (10.75%) cases with where collaterals grading was not assessed; 3 (3.22%) with missing mode of transfer; and 12 (12.90%) with missing door-in-door-out times.

The correlation matrix is shown in Figure 2. The highest correlation to receiving EVT was given by the occlusion location followed by onset to first CT. In other words, shorter times from onset to CT were more favourable for transfer, and occlusion locations in the ICA and M1 vessels rather than M2 or MCA resulted in a more favourable transfer. The feature importance analysis for decision tree and random forest is provided in Figure 3. The features that showed the greatest importance in both models were onset to first CT time and DIDO time. The occlusion location was important in both models but showed greater importance with decision tree. Conversely, age showed more importance with random forest. ASPECTS and mode of transfer showed greater importance with decision tree, while distance showed importance only with random forest.

The performance of all models is provided in Figure 4. Decision tree and random forest performed the best across all evaluation parameters. Decision tree had the best accuracy of 79.0%, an AUC of 79.0%, and precision of 80%. Random forest had an accuracy of 74%, an AUC of 75%, and a precision of 100%. The AUC curves are shown in Figure 5. SVM had the worst performance.

The resulting futile transfer rate (or those that the model indicated to transfer incorrectly) and the total false negative rate (the patient that should have been transferred out of the entire dataset) is shown in Figure 6. Random forest performed the best with a 0% futile transfer rate, and random forest has a false negative rate of 5.37%. Decision tree had a futile transfer rate of 18.86%, and it had the lowest false negative rate of 4.3%.

## 4. Discussion

This case study with a small number of patients indicates that ML can potentially lower futile transfer rates. The decision tree model and the random forest model both had good performance with an accuracy of 79% and 74%, respectively. The futile transfer rate for the decision tree model and random forest resulted in a 31.1% and 49.5% reduction in futile transfers, respectively, compared with physician decision alone; however, this comes as a cost of a 4.3% and 5.4% false negative rate, respectively. This shows that an ML model can potentially provide decision support to physicians making EVT transfer decisions for acute ischemic stroke patients.

The correlation matrix shows the clot location, collateral status, and onset to CT time have the greatest impact on whether EVT is received for transferred patients. Similarly, the feature importance for the random forest and decision tree reveals that in addition to the three from the correlation matrix, DIDO and age also impact whether a transferred patient will receive EVT. These results are reasonable, as time is critical for EVT eligibility and infarct will grow over time, so onset to CT remains a significant predictor to receiving EVT, as will the speed of transfer measured by DIDO. However, collateral status in combination with clot location is a modifier to EVT eligibility, as these will provide physiological pathways to keep the brain alive during transport. These findings match the inclusion criteria for the EVT trials, which ensured that patients had a large vessel occlusion and good collaterals [7,8].

The feature importance analysis allows us to provide a model explanation for the contribution of each feature. Both of these models show that the time from onset to first CT is most important along with the occlusion location. A review of the data with the feature importance information shows us that ICA and M1 occlusion locations and onset to first CT time of less than 100 min tends to result in a favourable transfer. This value was obtained by reviewing the onset to CT time data to determine at which point the transfers were more likely to be futile; however, this value may be different for different geographies with variable transfer distances, and it should not be viewed as a threshold for transfer. For longer onset to CT times, the patient should have good collateral status and a high ASPECTS score. Furthermore, the important of shorter DIDO times indicate that all thrombolysis centres should be working to reduce their transfer times to ensure favourable transfers for EVT.

There were several important features that would likely affect the decision to transfer, which could not be included in this study due to data availability. These features include stroke severity measured using the NIHSS (National Institute of Health Stroke Scale) and patient comorbidities. These variables are typically available to physicians during the time of transfer decision and may likely impact whether the transfer was futile. Future studies that look at the application of ML to provide a model to assist with decision support should include NIHSS and patient comorbidities.

There are likely missed patients in the real world as well who should have been transferred for EVT (false negatives). This dataset does not account for potential patients that should have been transferred for EVT but were not transferred. False negatives were not included in the dataset, because it is difficult to assess which patients would have received the EVT procedure if they had been transferred, as it is difficult to predict their status upon arrival at the EVT center. In actual clinical practice, it is also difficult to ascertain the number of false negatives as it is difficult to determine if these patients would have been eligible for EVT upon arrival, and it is highly dependent on physician’s practice and the ability of a health system to urgently transfer these patients. Future studies should include patients who were not transferred.

The decision to treat patients with EVT is highly physician-dependent [39]. Some physicians are more aggressive with treating patients that are outside of known guidelines (e.g., low ASPECTS at the EVT center), while other physicians may be more conservative. Therefore, an ML model cannot account for these types of variations, but it may be well suited to provide decision support to physicians when making transfer decisions.

The outcome measure of receiving EVT for transferred patients has some limitations. The decision to treat a transferred patient with EVT is affected by information that is not available at the time of transfer, as additional information is obtained after arrival at the EVT centre. Examples of this include the resolution of the stroke with thrombolysis, evolution to a symptomatic intracranial hemorrhage, and additional information obtained about the patient’s pre-morbid status.

This study uses a small dataset of only 93 patients, and the results may shift significantly with changes to guidelines or other changes. This is partly because of the small population of Nova Scotia, resulting in fewer patients that were transferred. Therefore, the results of this study can only provide a signal of whether ML can be used to develop software models to assist physicians with EVT transfer decisions. It is too early to use the random forest or decision tree model from this study to develop a decision support software application. Future studies with larger datasets are needed to further validate this study and to develop a more robust model that can then be used in a decision support software application. Additionally, this data is limited to a single health system with a single EVT center. The generalizability of these results to other health systems that may have multiple EVT centers is unknown. It is difficult to comment on how this data would transfer to other health systems and geographies, but some initial thoughts are that this model may be most relevant to single-payer health systems with a single urban EVT centre and multiple rural centres.

## 5. Conclusions

The decision tree and random forest ML models resulted in 31–49% fewer futile transfers with a cost of 4.3–5.4% false negatives. ML models can potentially be used to develop decision support software applications for EVT transfer decision making. However, future studies are needed with larger datasets and in different health systems to create a more robust model that can be generalized to other health systems.

## Figures and Tables

**Figure 1 healthcare-13-01435-f001:**
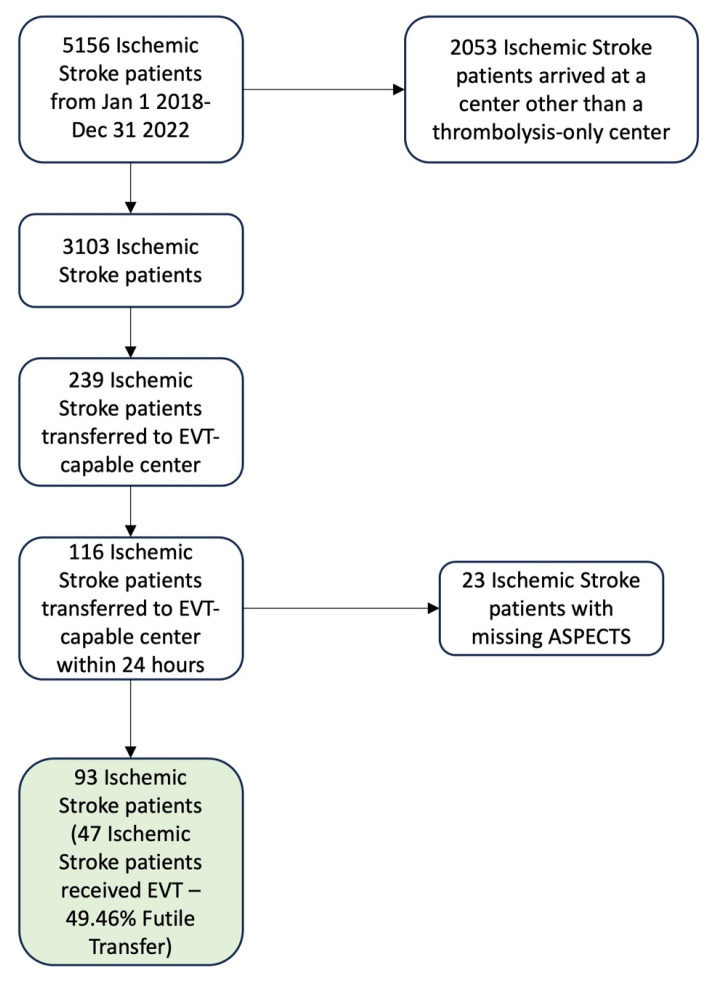
Total data inclusion and exclusion flowchart.

**Figure 2 healthcare-13-01435-f002:**
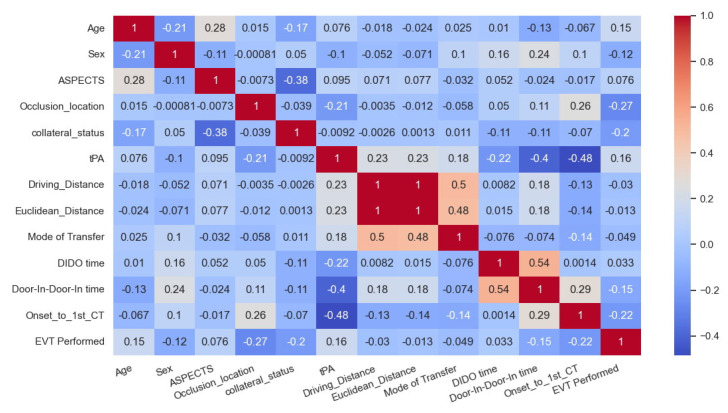
Correlation matrix for the 93-record dataset. EVT: endovascular thrombectomy, tPA: tissue plasminogen activator, CT: computed tomography.

**Figure 3 healthcare-13-01435-f003:**
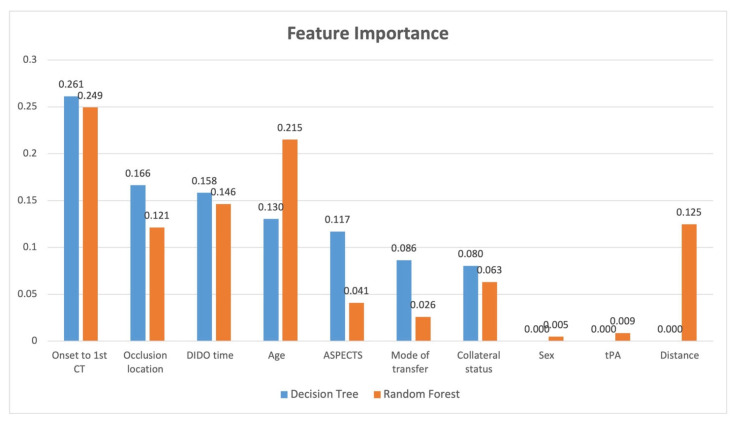
Feature importance analysis for decision tree and random forest. EVT: Endovascular thrombectomy, DIDO: Door-In-Door-Out, tPA: tissue plasminogen activator, CT: computed tomography.

**Figure 4 healthcare-13-01435-f004:**
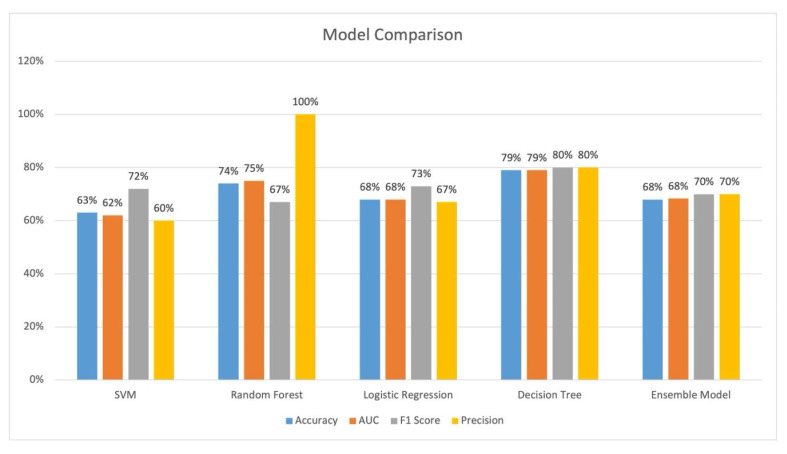
Model performance for the 93-record dataset. Legend: SVM: Support Vector Machine, ROC_AUC: Receiver Operating Characteristics Area Under the Curve.

**Figure 5 healthcare-13-01435-f005:**
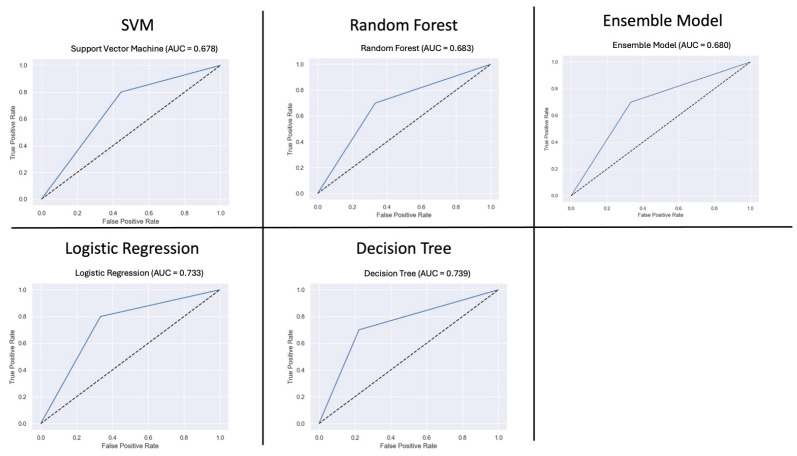
Area Under the Curve results for all 5 ML models. Legend: SVM: Support Vector Machine.

**Figure 6 healthcare-13-01435-f006:**
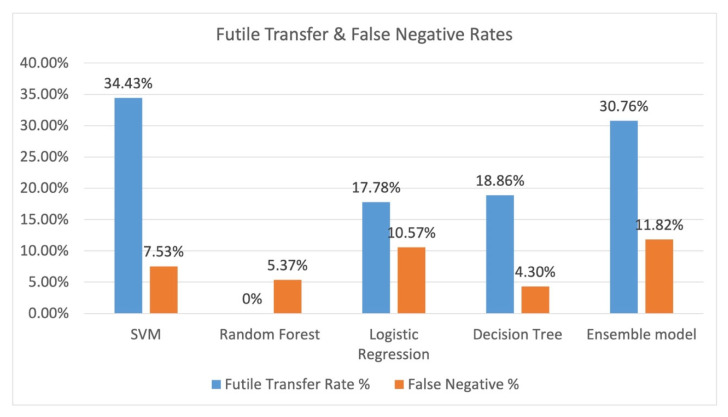
Futile transfer and false negative rates resulting from each model. Legend: SVM: Support Vector Machine.

**Table 1 healthcare-13-01435-t001:** List of all variables used in this study with the type of variable and their description. Legend: CT: Computed Tomography; ASPECTS: Alberta Stroke Program Early CT Score; CTA: CT Angiogram; ICA: Internal Carotid Artery; M1: Sphenoidal/Horizontal Segment; M2: Insular Segment; MCA: Middle Cerebral Artery; EVT: Endovascular Thrombectomy; km: kilometer.

Variable	Input/Output	Type	Description
Age	Input	Numeric	The age of the patient at the time of stroke onset (years).
Sex	Input	Categorical	The sex of the patient; converted to a binary variable.
Onset to first CT	Input	Numeric	The time from onset of stroke symptoms to the time of the CT scan at the thrombolysis-only center (minutes).
ASPECTS	Input	Integer (0–6)	The number of ischemic changes measured using ASPECTS at the thrombolysis-only center.
Clot Location	Input	Categorical	The thrombus location at the thrombolysis-only center based on the CTA; the following categories were used: tandem (0), terminal ICA (1), M1 (2), M2 (4), MCA (5) Other (7). Note: posterior was assigned a value of 3 but was subsequently removed.
Collateral Status	Input	Categorical	The level of collateral circulation using the following categories: good (0), intermediate (1), poor (2).
Thrombolysis	Input	Binary	Whether the patient received thrombolysis at the thrombolysis-only center: not given (0), given (1).
Distance	Input	Numeric	The driving distance between the thrombolysis center where the patient was first seen to the EVT-capable center. The Euclidean distance is a straight-line distance between the thrombolysis center where the patient was first seen to the EVT-capable center.Driving distance is used when ground transportation is used, and Euclidean distance is used when helicopter is used. (km)
Door-In-Door-Out	Input	Numeric	The time from the patient’s arrival at the thrombolysis-only center to the time of departure from this center.
Transfer Modality	Input	Categorical	The modality of transfer: ground (0), helicopter (1)
EVT Performed	Output	Binary	Whether EVT was performed (1) or not performed (0).

**Table 2 healthcare-13-01435-t002:** Summary descriptive statistics of entire dataset. Abbreviations: SD: Standard Deviation, IQR: Interquartile range, EVT: Endovascular thrombectomy, DIDO: Door-In-Door-Out, tPA: tissue plasminogen activator, CT: computed tomography.

Variable	Entire Dataset (n = 93)	Received EVT (n = 47)	Did Not Receive EVT (n = 46)
Age, Mean (±SD)	66.8 (±15.68)	69.11 (±12.47)	64.43 (±18.23)
Sex, Women (%)	47.31%	53.19%	41.30%
ASPECTS, Median (IQR)	9.0 (8.0–10.0)	9.0 (8.0–10.0)	9.0 (8.0–10.0)
Occlusion Location			
M1, %	42.52%	51.44%	33.60%
M2, %	17.24%	14.38%	20.07%
MCA, %	8.04%	0%	16.08%
Tandem, %	17.24%	20.29%	14.19%
Terminal ICA, %	13.79%	13.89%	13.69%
Other, %	1.17%	0%	2.37%
Collateral Status			
Good, %	61.45%	63.82%	59.08%
Intermediate, %	28.92%	31.93%	25.82%
Poor, %	9.63%	4.25%	15.01%
tPA, % that received	55.91%	63.82%	47.82%
Distance, median (IQR)	105.0 (87.40–195.0)	105.0 (92.70–157.0)	105.0 (8.0–221.7)
Mode of Transfer			
Ambulance, %	75.56%	78.72%	72.74%
Air, %	24.44%	21.28%	27.26%
DIDO Time, Median (IQR)	165.58 (114.18–243.47)	156.83 (108.07–211.0)	179.66 (123.43–319.92)
Onset to 1st CT, Median (IQR)	139.0 (69.50–230.50)	87.50 (57.75–196.25)	164.0 (99.0–366.0)

## Data Availability

The data presented in this study are not available due to privacy.

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
