# Peer review of "Predicting Ischemic Stroke Patients to Transfer for Endovascular Thrombectomy Using Machine Learning: A Case Study"

_healthcare, 2025, doi:10.3390/healthcare13121435_

Round 1

Reviewer 1 Report

Comments and Suggestions for Authors

Figure 3 is not clear kindly replace it better version

Stroke severity was excluded due to missing data is a major limitation

Not considered data related to comorbidities, imaging quality, or progression between imaging and treatment

The final dataset includes only 93 patients this limits the generalizability and robustness of the findings

The study finds decision tree and random forest models most effective, with significant reductions in futile transfer rates and relatively low false negative rates.

Summary Report:

This article presents a case study evaluating the application of machine learning to improve decision-making in transferring ischemic stroke patients for endovascular thrombectomy in Nova Scotia, Canada. Given the resource constraints and geographical barriers inherent in many health systems, this is a timely and required area of inquiry. The authors investigate whether ML algorithms can reduce futile inter-hospital transfers, thereby optimizing EVT utilization. The study utilizes data from the provincial stroke registry (2018–2022), applying four ML models (logistic regression, decision tree, random forest, and SVM), plus an ensemble model. Of 5156 stroke patients, only 93 patients met the final inclusion criteria. The study finds decision tree and random forest models most effective, with significant reductions in futile transfer rates and relatively low false negative rates.

This is a timely and clinically meaningful study that addresses a real-world inefficiency in stroke care using pragmatic machine learning tools. Despite a small sample size and some inherent limitations, the results are promising decision trees and random forests reduced futile transfers by up to 49.5% with modest false negative rates. This study offers a valuable proof-of-concept for using ML to guide critical transfer decisions in time-sensitive stroke management.

This study provides promising early evidence that ML models—particularly decision tree and random forest classifiers—may reduce futile EVT transfers without substantially increasing false negatives. The integration of clinically relevant and accessible variables is a strength, and the methodological rigor is commendable.

However, due to the small sample size and geographic specificity, these findings should be considered preliminary. The authors rightly caution against deploying the current models in clinical settings without further validation.

Overall, the article represents a thoughtful and well-executed pilot study that lays a foundation for future work in data-driven decision support for stroke care.

Author Response

Comment - Figure 3 is not clear kindly replace it better version
Response - Apologies for this, we have not replaced figure 3 with a more readable and higher resolution image.

Comment - Stroke severity was excluded due to missing data is a major limitation
Response - Thank you for this feedback. We agree that this is a major limitation.  Please see my response and revision to the text below.

Not considered data related to comorbidities, imaging quality, or progression between imaging and treatment
Thank you for this important comment.  We agree that comorbidities (as well as NIHSS) are important features that may have affected the model.  For this reason, we removed the brief sentence in the final paragraph, “Furthermore, the NIHSS variable could not be used in our model due to a large number of missing values, which is potentially a strong predictor of patients to transfer for EVT. A future study should include this variable in the dataset.”

Instead, we added the following paragraph earlier in the discussion (3rd paragraph of the discussion):
“There were several important features that would likely affect the decision to transfer, which could not be included in this study due to data availability.  These features include stroke severity measured using the NIHSS (National Institute of Health Stroke Scale) and patient comorbidities. These variables are typically available to physicians during the time of transfer decision, and may likely impact whether the was futile.  Future studies that look at the application of ML to provide a model to assist with decision support should include NIHSS and patient comorbidities. “

Since ASPECTS was required variable, we did not include any patients where the imaging quality was poor; however, this was not an issue in our dataset.  Furthermore, we did not include ASPECTS at the EVT-centre since this would not be available at the time of transfer.  

Comment - The final dataset includes only 93 patients this limits the generalizability and robustness of the findings
Response - We completely agree that this is a significant limitation to our study.  This is primary reason that we are calling this a case study, where we are looking at feasibility of ML to improve transfer decisions.  This is reason, we have stated the following, “This study uses a small dataset of only 93 patients, and the results may shift significantly with changes to guidelines or other changes.  This is partly because of the small population of Nova Scotia resulting in fewer patients that were transferred. Therefore, the results of this study can only provide a signal of whether ML can be used to develop software models to assist physicians with EVT transfer decisions. It is too early to use the Decision Tree model from this study to develop a decision support software application. Future studies with larger datasets are needed to further validate this study and to develop a more robust model that can then be used in a decision support software application.”

Comment - The study finds decision tree and random forest models most effective, with significant reductions in futile transfer rates and relatively low false negative rates.

Summary Report:
This article presents a case study evaluating the application of machine learning to improve decision-making in transferring ischemic stroke patients for endovascular thrombectomy in Nova Scotia, Canada. Given the resource constraints and geographical barriers inherent in many health systems, this is a timely and required area of inquiry. The authors investigate whether ML algorithms can reduce futile inter-hospital transfers, thereby optimizing EVT utilization. The study utilizes data from the provincial stroke registry (2018–2022), applying four ML models (logistic regression, decision tree, random forest, and SVM), plus an ensemble model. Of 5156 stroke patients, only 93 patients met the final inclusion criteria. The study finds decision tree and random forest models most effective, with significant reductions in futile transfer rates and relatively low false negative rates.
This is a timely and clinically meaningful study that addresses a real-world inefficiency in stroke care using pragmatic machine learning tools. Despite a small sample size and some inherent limitations, the results are promising decision trees and random forests reduced futile transfers by up to 49.5% with modest false negative rates. This study offers a valuable proof-of-concept for using ML to guide critical transfer decisions in time-sensitive stroke management.
This study provides promising early evidence that ML models—particularly decision tree and random forest classifiers—may reduce futile EVT transfers without substantially increasing false negatives. The integration of clinically relevant and accessible variables is a strength, and the methodological rigor is commendable.
However, due to the small sample size and geographic specificity, these findings should be considered preliminary. The authors rightly caution against deploying the current models in clinical settings without further validation.
Overall, the article represents a thoughtful and well-executed pilot study that lays a foundation for future work in data-driven decision support for stroke care.
Response - Thank you for this precise and thoughtful summary.  We agree with all the points, and we hope that this manuscript can spark further research into the use of ML models to assist with EVT transfer decision support for ischemic stroke patients.

Reviewer 2 Report

Comments and Suggestions for Authors
  • The authors should clearly and specifically state the objective of the study in the abstract.
  • There are unnecessary extra spaces between words in the abstract; these should be removed.
  • In the Methods section of the abstract, the authors should indicate the type of study conducted (e.g., retrospective study).
  • Keywords should be listed in alphabetical order (A to Z).
  • The authors are advised to include a graphical abstract that visually represents the study's methodology and key findings.
  • In line 45, please include the estimated number of deaths caused by stroke, the prevalence of acute ischemic stroke, and relevant data on associated morbidity.
  • In lines 45–46, the authors should elaborate on how the introduction of thrombolysis has transformed clinical practice and contributed to improved patient outcomes, including reductions in mortality and morbidity over time.
  • For the statements made in lines 46–48, a supporting reference should be added.
  • In line 46, replace the word "let" with a more appropriate alternative.
  • In line 51, references 3 and 4 should be placed before the period. Ensure that all in-text citations throughout the manuscript are consistently positioned before the full stop.
  • Please check whether a more recent meta-analysis on the efficacy of endovascular treatment (EVT) has been published. If so, include the relevant data and citation around line 56.
  • The sentence, "This results in a NNT (number needed to treat) for a reduction in disability of 2.6," is unclear. Please rephrase for clarity.
  • In line 56, the abbreviation should be formatted as: NNT (Number Needed to Treat).
  • In line 67, replace the phrase “In other words” with a more formal alternative.
  • Also in line 67, the expression “there is great waste of…” should be rephrased using more formal language.
  • The meaning of the phrase “This essentially uses the philosophy of ‘casting a wide net’…” is unclear. Please elaborate or consider rewording.
  • The introduction currently lacks a clear rationale for choosing EVT over thrombolysis. The authors should include comparative data to highlight the advantages of EVT.
  • In line 80, the authors should expand on how machine learning (ML) can benefit healthcare systems and patients more broadly
  • In lines 101 and 102, the comparisons regarding size (e.g., “similar to the European country of Slovakia” and “half the size of the US state of Virginia”) should be removed from the main manuscript, along with the map presented in Figure 1. This information is better suited for inclusion in the supplementary materials.
  • The authors should clarify why the NIHSS (National Institutes of Health Stroke Scale) was not included among the variables analyzed.
  • Please explain the rationale behind calculating only the false negative rate. Why were false positives or other error metrics not considered?
  • The authors should specify which statistical tests were used to assess the normality of the data.
  • Mean values should be reported in the format: Mean (± Standard Deviation).
  • The table font should be consistent with the rest of the manuscript text for uniformity.
  • All abbreviations used in the table should be defined in a footnote or at the end of the table.
  • The Discussion section is currently too brief and lacks depth. It should be expanded and restructured to provide a more comprehensive analysis and interpretation of the findings.

Author Response

 Comment ·         The authors should clearly and specifically state the objective of the study in the abstract.
Response - Thank you for alerting us to this.  We have added the following to the abstract: “Objective: The objective of the study is to determine if ML can provide more accurate selection of patients to transfer for EVT.”
Comment ·         There are unnecessary extra spaces between words in the abstract; these should be removed.
Response - Apologies for this.  The extra spaces have been removed.
Comment·         In the Methods section of the abstract, the authors should indicate the type of study conducted (e.g., retrospective study).
Response - The type of study has been added to the methods section of the abstract.
Comment·         Keywords should be listed in alphabetical order (A to Z).
Response - The keywords have now been listed in alphabetical order.
Comment·         The authors are advised to include a graphical abstract that visually represents the study's methodology and key findings.
Thank you for this suggestion. A graphical abstract was added.
Comment·         In line 45, please include the estimated number of deaths caused by stroke, the prevalence of acute ischemic stroke, and relevant data on associated morbidity.
Thank you for this suggestion.  The following text was added:
“Data from Nova Scotia shows that there are 125 ischemic stroke per 100,000 people in 2017 [2], and national data shows the morality due to ischemic stroke is 12.1% with 34.6% discharged home without services, 11.4% discharged home with services, 28.4% discharged to inpatient rehabilitation, and remainder discharged to long-term care [3]”
The following 2 references were added to support the data above:
1.    Holodinsky, J.K., Lindsay, P., Yu, A.Y.X, Ganesh, A., Joundi, R.A., Hill, M.D. Estimating the number of hospital or emergency department presentations for stroke in Canada. Canadian Journal of Neurological Sciences 2023, 50:820-825.
2.    Kamal, N., Lindsay, M.P., Côté R., Fang, J., Kapral, M.K., Hill, M.D. Ten-year trends in stroke admission and outcomes in Canada. Canadian Journal of Neurological Sciences 2023, 42:168-175.

Comment·         In lines 45–46, the authors should elaborate on how the introduction of thrombolysis has transformed clinical practice and contributed to improved patient outcomes, including reductions in mortality and morbidity over time.
Response - Thank you for this great suggestion.  I have elaborated on the evidence and added the following to the original sentence:
“Fortunately, thrombolysis treatment for acute ischemic stroke was proven in 1995 [4] and the subsequent pooled analysis [5] showed the criticality of fast treatment and the need to treat with thrombolysis within 4.5 hours with the odds of excellent outcomes with thrombolysis at 1.75 (1.35-2.27) when treated in less than 3 hours, and this falls to 1.15 (0.95-1.40) when treated after 4.5 hours. Similarly, the mortality also increases with treatment delays from 1.00 (0.81-1.24) in less than 3 hours to 1.22 (0.99-1.50) when treated after 4.5 hours.[5]”
The following reference was added:
Emberson, J., Lee, K.R., Lynden, P., Blackwell, L, Albers, G., Bluhmki, E., Brott, T., Cohen, G., Davis, S., Donnan, G. Grotta, J. et al. effect of treatment delay, age, and stroke severity on the effects of intravenous thrombolysis with aleplase for actue ischaemic stroke: a meta-analysis of individual patient data from randomised trials. Lancet 2014,384:1929-1935. 

Comment·         For the statements made in lines 46–48, a supporting reference should be added.
Response - Thank you for making this suggestion.  I have added the reference (#6) to support this, which are the current Canadian Stroke Best practice publication.
Comment·         In line 46, replace the word "let" with a more appropriate alternative.
Response - We have replaced the word “led” to “guided”.

Comment·         In line 51, references 3 and 4 should be placed before the period. Ensure that all in-text citations throughout the manuscript are consistently positioned before the full stop.
Response - Thank you for this clarification.  This has been updated throughout the manuscript.
Comment·         Please check whether a more recent meta-analysis on the efficacy of endovascular treatment (EVT) has been published. If so, include the relevant data and citation around line 56.
Response - These are the 5 main EVT trials that are consistently used on the stroke literature.  The meta analysis was done as part of the HERMES Consortium.  The two key publications have been cited as 7 and 10.
Comment·         The sentence, "This results in a NNT (number needed to treat) for a reduction in disability of 2.6," is unclear. Please rephrase for clarity.
Response - Apologies for the lack of clarity.  We have updated this sentence to now read, “The pooled analysis demonstrated that EVT is a highly efficacious treatment with a NNT (number needed to treat) of 2.6 for improving a reduction in disability, or improving the modified Rankin scale score by 1 point at 90 days [7].”
Comment·         In line 56, the abbreviation should be formatted as: NNT (Number Needed to Treat).
Response - This has now been updated.
Comment·         In line 67, replace the phrase “In other words” with a more formal alternative
Response: We have updated this to “Specifically”.
Comment·         Also in line 67, the expression “there is great waste of…” should be rephrased using more formal language.
Response - This sentence now reads, “Specifically, the futile transfers forces the use of scarce ambulance and air transport resources for no benefit to the patient with 66% of those transferred for EVT being deemed ineligible for the treatment upon arrival [12].”
Comment·         The meaning of the phrase “This essentially uses the philosophy of ‘casting a wide net’…” is unclear. Please elaborate or consider rewording.
Response - We added the following explanatory statement, “which means that the stroke system transfers patients with even a small change of receiving treatment”
Comment·         The introduction currently lacks a clear rationale for choosing EVT over thrombolysis. The authors should include comparative data to highlight the advantages of EVT.
Response - In Stroke treatment, thrombolysis and EVT are given together.  They are synergistic treatments, and EVT is not preferred over thrombolysis.  In the EVT trials, 75% of EVT patients received thrombolysis.  The other 25% were those who were not eligible for thrombolysis treatment, which could be arrival beyond the 4.5 hour thrombolysis treatment window or anti-coagulant use.  We have added the following statement after the description of the EVT trial results, “Thrombolysis treatment is synergistic to EVT, and 75% of EVT patients in the EVT trials received thrombolysis, and the other 25% were not eligible to receive thrombolysis treatment.”
Comment·         In line 80, the authors should expand on how machine learning (ML) can benefit healthcare systems and patients more broadly
Response - Thank you for this suggestion.  We agree that this is helpful.  The following has been added to the introduction, “Machine learning (ML) ML is a branch of artificial intelligence that builds data-driven statistical models to predict specific outcomes, which can be very helpful in health care systems; for example, ML can be used to provide decision support to clinicians where uncertainty and ambiguity exists, thus the ML models can be developed for health care systems to optimize patient outcomes.”
Comment·         In lines 101 and 102, the comparisons regarding size (e.g., “similar to the European country of Slovakia” and “half the size of the US state of Virginia”) should be removed from the main manuscript, along with the map presented in Figure 1. This information is better suited for inclusion in the supplementary materials
Response - Thank you for this suggestion.  This text has been removed and the figure has been moved to the supplemental.
Comment·         The authors should clarify why the NIHSS (National Institutes of Health Stroke Scale) was not included among the variables analyzed
Response - Yes, we completely agree that NIHSS is an important variable, which we could not include due to data availability.  We added the following to clarify why it was not collected, “Unfortunately, we could not include NIHSS, due to this variable being missing in the majority of records. The collection of NIHSS has been challenging to collect at smaller regional centres, but the importance of this measure is recognized, and this will be a limitation of this current study”
Comment·         Please explain the rationale behind calculating only the false negative rate. Why were false positives or other error metrics not considered?
Response - Thank you for bringing this up.  We are reporting F1 Score and futile transfer rate, which is a measure that considers both fast positives and false negatives, thus false positives are considered in the determination of accuracy of the models. The reason for reporting on false negative separately is because if the model misses a patient to transfer for EVT, this will result is a poor outcome for the patient. However, it the model gives a false positive, there is no poor patient outcome but rather a higher cost to the health system for the unnecessary transfer. We have added the following text to address this, “The reason for a specific focus on false negative is because a false negative has the greatest negative impact in the use of the ML for decision-making; specifically, a false-negative in this model means that the model is recommending that a patient is not transferred for EVT when they are potentially eligible for this treatment, which means that this would result is a poor patient outcome. False positive on the other hand would not result is poor patient outcome, as transferring a patient who would not be eligible for EVT results in the same patient outcome, but at a higher cost to the healthcare system due to the unnecessary transfer. False positives are taken into consideration in the F1-Score and futile transfer rate”
Comment·         The authors should specify which statistical tests were used to assess the normality of the data.
Response - We added the following to the methods section, “Tests for normality were conducted using the Shapiro-Wilk test.”
Comment·         Mean values should be reported in the format: Mean (± Standard Deviation).
Response - We have added the ± to all areas where we reported mean (SD).
Comment·         The table font should be consistent with the rest of the manuscript text for uniformity.
Response - The table font has been updated to the manuscript font.
Comment·         All abbreviations used in the table should be defined in a footnote or at the end of the table.
Response - All abbreviations in the tables have been added as a footnote to the table
Comment·         The Discussion section is currently too brief and lacks depth. It should be expanded and restructured to provide a more comprehensive analysis and interpretation of the findings.
Response - Thank you for this recommendations.  The following two paragraphs have been added to the discussion:
“The correlation matrix shows the clot location, collateral status and onset to CT time have the greatest impact on whether EVT is received for transferred patients. Similarly, the feature importance for the random forest and decision tree reveal that in addition to the three from the correlation matrix DIDO and age also impact whether transferred patient will receive EVT. These results are reasonable as time is critical for EVT eligibility and infarct will grow over time, so onset to CT remains a significant predictor to receiving EVT as will the speed of transfer measured by DIDO. However, collateral status in combination with clot location is a modifier to EVT eligibility, as these will provide physiological pathways to keep the brain alive during transport. These finding match the inclusion criteria for the EVT trials, which ensured that patients had a large vessel occlusion and good collaterals [7,8].”
“There were several important features that would likely affect the decision to transfer, which could not be included in this study due to data availability. These features include stroke severity measured using the NIHSS (National Institute of Health Stroke Scale) and patient comorbidities. These variables are typically available to physicians during the time of transfer decision, and may likely impact whether the was futile.  Future studies that look at the application of ML to provide a model to assist with decision support should include NIHSS and patient comorbidities.” 

Reviewer 3 Report

Comments and Suggestions for Authors

The present study, “Predicting Ischemic Stroke Patients to Transfer for Endovascular Thrombectomy Using Machine Learning,” is compelling; however, several points need to be addressed during revision.

  1. In the introduction, the transition to the paragraph discussing machine learning (ML) is somewhat abrupt. It moves directly from challenges in patient transfer and resource utilization to ML applications without a clear connecting statement or rationale.
  2. Why was the decision tree model preferred in the conclusion, even though the random forest model had a lower futile transfer rate (0%) as stated in the abstract? This discrepancy needs clarification.
  3. The rationale for choosing the thresholds (e.g., <100 minutes onset-to-CT) isn't explained. Are these thresholds data-driven, clinical, or arbitrarily chosen? Please clarify how they were derived and whether this information was used to train or test the model.
  4. Can you elaborate on how these false negatives compare to current clinical practice? Are these rates clinically acceptable, and how might they affect decision-making?
  5. Are there any preliminary insights into how transferable this model might be to regions with different EVT logistics (e.g., urban vs. rural, multiple EVT centres)?
  6. In Figure 3, the correlation matrix image is unclear. Provide a higher-resolution or clearer version for better readability.
  7. In Figures 4, 5, and 7, label the parameter on the Y-axis to improve clarity and facilitate interpretation.
  8. Line 241, “Random Rorest” change to “Random Forest”.

Author Response

The present study, “Predicting Ischemic Stroke Patients to Transfer for Endovascular Thrombectomy Using Machine Learning,” is compelling; however, several points need to be addressed during revision.
Comment 1.    In the introduction, the transition to the paragraph discussing machine learning (ML) is somewhat abrupt. It moves directly from challenges in patient transfer and resource utilization to ML applications without a clear connecting statement or rationale.
Response - Thank you for bringing this to our attention.  We have now added the following introductory sentence to the beginning of ML paragraph in to introduction, “The challenges with decision to transport acute ischemic stroke patients for EVT can potentially be aided with the application of Machine learning (ML).”
Comment 2.    Why was the decision tree model preferred in the conclusion, even though the random forest model had a lower futile transfer rate (0%) as stated in the abstract? This discrepancy needs clarification.
Response - Thank you for bringing this to our attention.  The conclusion was meant to include both Decision Tree and Random Forest, as the data for both models were included.  We have now updated the test to include both models.
Comment 3.    The rationale for choosing the thresholds (e.g., <100 minutes onset-to-CT) isn't explained. Are these thresholds data-driven, clinical, or arbitrarily chosen? Please clarify how they were derived and whether this information was used to train or test the model.
Response - This value was not a threshold, but simple the onset-to-CT time value where any longer times tended to result in futile transfers.  We have added the following to the discussion to clarify this value, “This can potentially values was obtained by reviewing the onset to CT time data to determine at which value the transfers were more likely to be futile; however, this value may be different for different geographies with variable transfer distances, and it should not be viewed as a threshold for transfer”
Comment 4.    Can you elaborate on how these false negatives compare to current clinical practice? Are these rates clinically acceptable, and how might they affect decision-making?
Response - It is really hard to determine the number of false negatives in current clinical practice, as patients that are not transferred for EVT are typically not tracked, and there is no way to ascertain if these patients would have been eligible for EVT if they had been transferred.  This is highly dependent on the physician’s practice and the health system’s ability to urgently transfer these patients.  The following was added to this section of the discussion,” In actual clinical practice, it is also difficult to ascertain the number of false negatives as it is difficult to determine if these patients would have been eligible for EVT upon arrival, and it is highly dependent on physician’s practice and the ability of a health system to urgently transfer these patients.”

Comment 5.     Are there any preliminary insights into how transferable this model might be to regions with different EVT logistics (e.g., urban vs. rural, multiple EVT centres)?
Response - Thank you for bring this up, as it is important to consider and comment on.  The following is added to the final paragraph of the discussion, “It is difficult to comment on how this data would transfer to other health systems and geographies, but some initial thoughts are that this model may be most relevant to single-payer health systems with a single urban EVT centre and multiple rural centres”

Comment 6.     In Figure 3, the correlation matrix image is unclear. Provide a higher-resolution or clearer version for better readability.
Response - This image has now been updated to a better quality image. Thank you.
Comment 7.     In Figures 4, 5, and 7, label the parameter on the Y-axis to improve clarity and facilitate interpretation.
Response - The y-axis is given by a percentage for that output measure.
Comment 8.     Line 241, “Random Rorest” change to “Random Forest”.
Response - This has now been corrected.  Apologies for this.

Round 2

Reviewer 2 Report

Comments and Suggestions for Authors

Most of my concrens have been addressed.

Author Response

Thank you for reviewing.  Your comments improved this manuscript. We have revised the manuscript for English and final polishing.  

Reviewer 3 Report

Comments and Suggestions for Authors

In Figure 5, the AUC values are shown with inconsistent decimal precision—some extend beyond ten digits after the decimal point, while the Ensemble model displays only two. For clarity and consistency, I recommend standardizing all AUC values to three decimal places.

Author Response

Thank you for your comments and review.  I have now revised Figure 5 to only have 3 decimal points after the AUC.  Thank you for finding this inconsistency, and bringing it to my attention.

I have also revised the English throughout.